# Successful treatment of recurrent visceral leishmaniasis relapse in an immunocompetent adult female with functional hypopituitarism in Bangladesh

Tabiha Binte Hannan[1]*, Zazeba Hossain[1], Utshab Roy[1], S. M. Mahbubur Rahman[1], Md Sadiqur Rahman[1], Sadia Sabah[1], Md. Abu Rahat[2], Rajashree Chowdhury[2], Faria Hossain[2], Dinesh Mondal[2], Shampa Saha[3], Tahniyah Haq[4], Md Rafiqul Alam[1], Fazle Rabbi Chowdhury[1]

**1** Department of Internal Medicine, Bangabandhu Sheikh Mujib Medical University (BSMMU), Dhaka, Bangladesh, **2** Nutrition Research Division, International Centre for Diarrhoeal Disease Research (icddrb), Dhaka, Bangladesh, **3** Bangladesh Society of Infectious and Tropical Diseases (BSITD), Dhaka, Bangladesh, **4** Department of Endocrinology, Bangabandhu Sheikh Mujib Medical University (BSMMU), Dhaka, Bangladesh

\* tabiha.44@gmail.com

**Data Availability Statement:** Data available at 10.6084/m9.figshare.25467478.

## Abstract

### Background

Currently available treatment options are mostly effective in achieving long-term cure in visceral leishmaniasis (VL) patients. However, there have been reports of recurrence of this illness in both immunosuppressed and immunocompetent patients.

### Case presentation

We report the first case of recurrent VL relapse in a 19-year-old immunocompetent female with functional hypopituitarism (hypogonadotropic hypogonadism with central hypothyroidism) from Bangladesh, who has been treated three times previously with optimal dosage and duration- liposomal amphotericin B (LAmB) alone and in combination with miltefosine. We treated the patient successfully with a modified treatment regimen of 10 mg/kg body weight LAmB for two consecutive days along with oral miltefosine for seven days as loading dose. For secondary prophylaxis, the patient received 3 mg/kg body weight LAmB along with oral miltefosine for seven days monthly for five doses followed by hormonal replacement. The patient remained relapse free after 12 months of her treatment completion.

### Conclusion

In the absence of protective vaccines against *Leishmania* species and standard treatment regimen, this modified treatment regimen could help the management of recurrent relapse cases.

**Funding:** The author(s) received no specific funding for this work.

**Competing interests:** The authors have declared that no competing interests exist.

## Author summary

Visceral leishmaniasis is a globally recognized neglected tropical disease, and Bangladesh is a major contributor of the global burden of this illness. In this article, we present a case on a 19-year-old immunocompetent female with pubertal delay, who has been diagnosed as a case of visceral leishmaniasis for 11 years and presented to us with recurrent relapses despite being treated with standard treatment regimens. The patient also developed functional hypopituitarism due to this chronic illness, which hindered her personal, familial, as well as social life. In scarcity of evidence of a standard treatment regimen for recurrent relapse cases, we treated her with high dose liposomal amphotericin B and miltefosine, along with secondary prophylaxis combining these two drugs with excellent result. Her pubertal delay was also addressed with utmost importance to re-establish her social confidence. This article could encourage physicians to outline an effective regimen to deal with these difficult-to-treat cases.

## Introduction

In 2021, a total of 11,743 cases of visceral leishmaniasis (VL) have been reported from WHO region. It is an endemic disease in four eco-epidemiological regions- Americas, East Africa, North Africa, West and South-east Asia. 1464 (13%) cases of new VL have been reported from South-east Asia in 2021 and 35 cases have been reported from Bangladesh with a case fatality rate of 6.9% [1, 2].

Several cases of frequent relapses of VL have been reported after optimal treatment regimen in immunosuppressed hosts (i.e., people living with HIV, patients with organ transplant etc.) around the world [3–5]. However, very few cases of recurrent VL in immunocompetent hosts have been reported to date [6]. Unfortunately, there is no definite treatment regimen available to treat these cases. To the best of our knowledge, here we report the first case of recurrent relapses of VL in an immunocompetent host with functional hypopituitarism from Bangladesh, who was treated successfully with a treatment regimen combining liposomal amphotericin B (LAmB) and miltefosine. We followed a treatment regimen reported by Lagadinou *et. al.*, who treated a case of recurrent relapses of visceral leishmaniasis in a Greek immunocompetent male [6].

## Case presentation

A 19-year-old female, hailing from Narsingdi, Bangladesh, was admitted into our hospital as a diagnosed case of recurrent visceral leishmaniasis relapse (VL) with the complaints of prolonged fever, weight loss, short stature and gradual swelling of abdomen. On background history, she had a long period of suffering due to her illness. She first developed irregular bouts of fever in 2012, along with progressive unintentional weight and gradual distension of abdomen. She sought medical consultation from local physicians, with no clinical improvement. In November 2018, she was first diagnosed as a case of new kala-azar on the basis of clinical presentations and positive rk-39 test in a district hospital. She was treated with single dose of 10 mg/kg body weight liposomal amphotericin-B (LAmB) with no improvement. In July 2019, she presented with high grade irregular fever, epistaxis, gum bleeding, weight loss, and was diagnosed as a case of relapse VL. She was treated with 15 mg/kg body weight LAmB in three divided doses (5 mg/kg body weight on every alternate day for three doses). Unfortunately, her symptoms did not improve this time either, and was re-admitted after two months. She was

**Table 1. Investigation summary of the patient.**

| Investigations | Before treatment | Loading dose (10 mg/kg BW LAmB+ Oral Miltefosine from day 2-day 8) | After 1st cycle | After 2nd cycle | After 3rd cycle | After 4th cycle | After 5th cycle | 6 months after treatment completion | 12 months after treatment completion | Normal range |
|---|---|---|---|---|---|---|---|---|---|---|
| | | | Secondary prophylaxis (3 mg/kg BW LAmB+ Oral miltefosine from day 2- day 8) | | | | | | | |
| Date | October 2021 | November 2021 | December 2021 | January 2022 | February 2022 | March 2022 | April 2022 | October 2022 | February 2023 | |
| Hb(gm/dL) | 6.99 | 9.2 | 9.6 | 10.7 | 8.4 | 10.5 | 11.5 | 11.2 | | 13.5–15.5 |
| ESR (mm in 1st hour) | 115 | 95 | 20 | 25 | 43 | 25 | 30 | 25 | | 0–10 |
| Total WBC count (/mm$^3$) | $0.68\times10^9$ | $2\times10^9$ | $3\times10^9$ | $2\times10^9$ | $3.5\times10^9$ | $3.5\times10^9$ | $4.5\times10^9$ | $4\times10^9$ | | $7\pm3\times10^9$ |
| Platelet count (/mm$^3$) | $10\times10^9$ | $40\times10^9$ | $80\times10^9$ | $75\times10^9$ | $120\times10^9$ | $76\times10^9$ | $100\times10^9$ | $100\times10^9$ | | $150-450\times10^9$ |
| rK-39 strip test | Positive | Positive | Positive | Positive | Positive | Positive | Positive | - | | |
| Buffy coat of blood for PCR for Leishmania DNA | Positive | Positive | Negative | Negative | Negative | Negative | Negative | Negative | | |

Hb: Hemoglobin, ESR: Erythrocyte sedimentation rate, WBC: White blood cells, HIV: Human Immunodeficiency Virus, MT: Mantoux test, PCR: Polymerase chain reaction,

tagged as a case of recurrent relapse of VL and was treated with single dose LAmB 5 mg/kg body weight, followed by miltefosine 60 mg in two divided doses for seven days with no clinical improvement thereafter. She was lost to follow-up between 2019–2021 and in October 2021, the patient was referred to our hospital by national kala azar elimination programme (NKEP) Bangladesh.

On admission, the patient was febrile, severely malnourished evidenced by sparse, thin hair, gross muscle wasting. Her height was 130 cm (below 5th percentile), weight 20 kg (below 5th percentile) and BMI 11.8 kg/m$^2$(below 5th percentile). She had hepatomegaly (12 cm), and massive splenomegaly (25 cm). Our patient was also amenorrheic, and there was no evidence of secondary sex characteristic development (Stage 1 breast development, and absence of axillary and pubic hair). We investigated her as much possible to exclude possible causes of immunosuppression. Her random blood sugar (RBS) and Chest X Ray were normal, and Anti-HIV (1+2), Mantoux test were also negative. Her serum albumin was low (22 gm/L). Her other investigation profile is summarized in Table 1.

The patient was treated with four units of fresh whole blood transfusion before starting anti-leishmania treatment. As she got multiple doses of the two most potent drugs available in our country, it was very challenging to establish a suitable treatment regimen for her. We treated our patient with a loading dose of high concentration LAmB (10 mg/kg body weight for two consecutive days) followed by oral miltefosine (2.5 mg/kg body weight daily) for seven days. To prevent the risk of recurrence, monthly secondary prophylaxis with LAmB at a dose of 3 mg/kg body weight followed by oral miltefosine (2.5 mg/kg body weight daily) for seven days for the next five months was planned. We also planned for a splenectomy if her thrombocytopenia and massive splenomegaly persisted after treatment. On her subsequent visits, clinical and laboratory improvements were monitored. To our relief, significant clinical improvement was noticed after the loading dose of LAmB evidenced by absence of fever, gradual weight gain, and decreasing size of liver and spleen. PCR for *Leishmania* DNA was done every month and it became negative after the 1st dose of secondary prophylaxis. During her

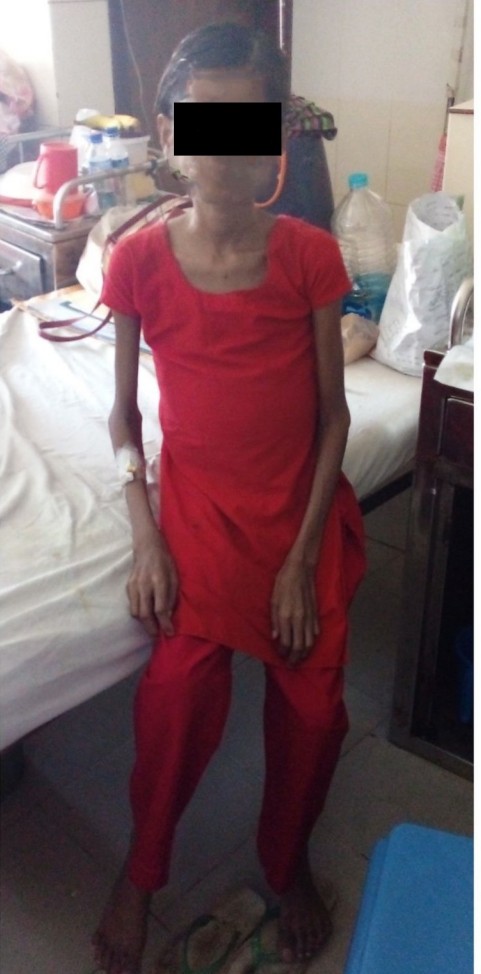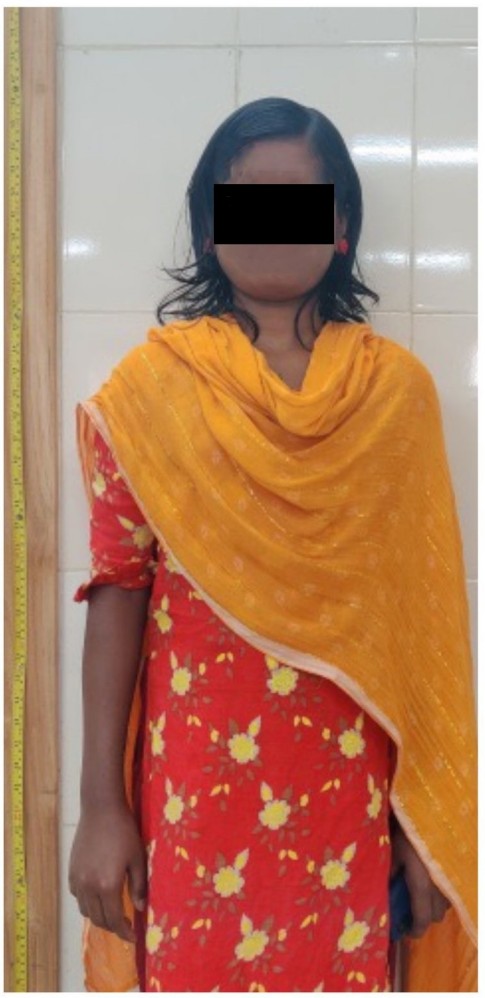

**Fig 1. Patient's physical improvement during and after treatment completion.** Patient's physical appearance during 2[nd] month of treatment showing gross muscle wasting, short stature, and abdominal distension and physical improvement after 6 months of treatment completion showing marked improvement in general physical appearance (N.B. picture taken by the first author and used with informed written consent from the patient).

course of treatment, she occasionally developed hypokalemia due to LAmB toxicity, which was treated with oral potassium chloride replacement. After six months of her treatment completion, she did not have any febrile episode, weight was 27 kg, she had splenomegaly (4 cm) but no hepatomegaly, platelet count was increased to 100,000/mm$^3$, rK-39 was positive, however PCR for *Leishmania* DNA was negative. Her physical improvements during and after treatment is depicted in Fig 1A and 1B.

Alongside her treatment for visceral leishmaniasis, she was evaluated thoroughly by the department of endocrinology for short stature and primary amenorrhea in June 2021. She had low serum estradiol, free T$_4$ along with normal serum LH, FSH, TSH (Table 2) and normal pituitary gland on MRI shown in Fig 2.

She was diagnosed as a case of functional panhypopituitarism (hypogonadotropic hypogonadism with central hypothyroidism). Her hormone profile is given in Table 1. She was treated with conjugated estrogen and levothyroxine 50 mcg. In October 2022, after six months

**Table 2. Endocrinological profile of the patient.**

| Investigation Name | Result/Interpretation | Normal range |
|---|---|---|
| Serum TSH (µIU/mL) | 5.7 | 0.34–5.6 |
| Serum FT$_4$ (ng/dL) | 0.49 | 0.61–1.61 |
| Serum FSH (IU/L) | 4.16 | 2.5–10.2 |
| Serum LH (IU/L) | 7.68 | 1.9–12.5 |
| Serum Estradiol (pg/mL) | 18.9 | 21–251 |
| Serum Cortisol (nmol/L) | 514.09 | 138–690 |
| USG of whole abdomen with special attention to genital organs | Length of uterus: 5.5 cm, Endometrial thickness: 6.8 mm | |
| MRI of Pituitary Gland | Normal study | |

LH: Luteinizing hormone, FSH: Follicle stimulating hormone, TSH: Thyroid stimulating hormone, FT$_4$: Free T$_4$

treatment with 0.3125 mg conjugated estrogen, her height increased to 140 cm, weight 27 kg, and she had stage 3 breast development. In her last follow-up visit in February 2023, after four months of treatment with 0.625 mg conjugated estrogen, her height was 146 cm, weight was 29 kg, and BMI was 13.6 kg/m$^2$. She had stage 4 breast development, and adequate uterus size

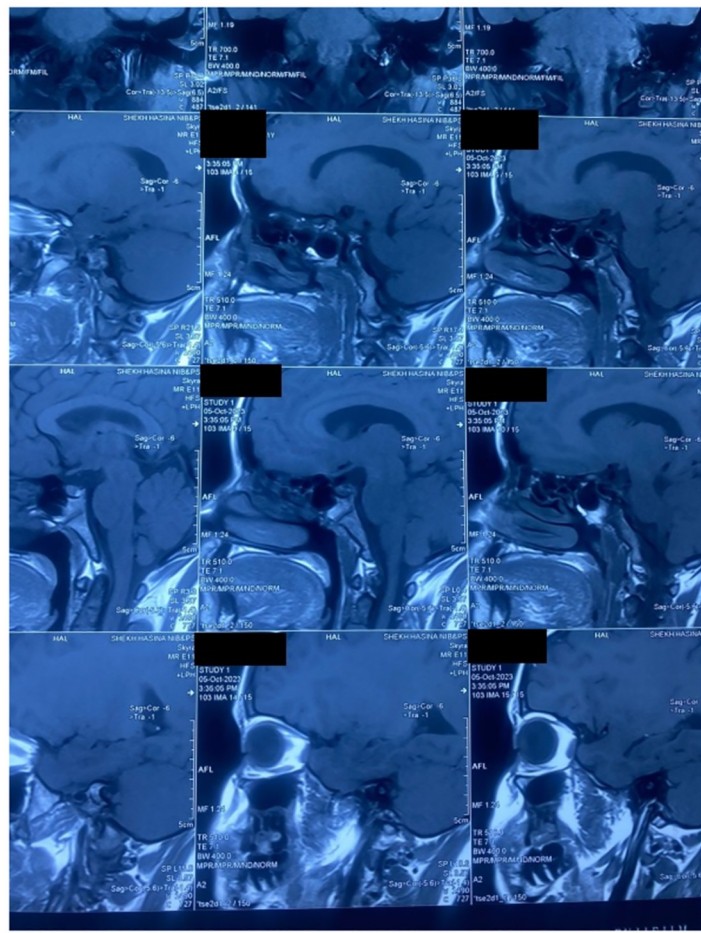

**Fig 2. MRI of pituitary gland of the patient.**

(Table 1). She was started on cyclical estrogen and progesterone and has been having regular withdrawal bleeds since then. After 18 months of hormone replacement therapy, estrogen and progesterone therapy was withhold to evaluate establishment of her natural menstrual cycle. During her most recent follow up in March 2024, she stated that she has been having regular menstrual cycle for the last 4 months.

## Discussion

*Leishmania* are internalized by host macrophages into liver, spleen, bone marrow and lymph nodes after the bite of an infected sandfly. This is why, it is a matter of challenge for the anti-leishmania drugs to access the intracellular organisms and result in parasite eradication [7]. Immunocompetent hosts can produce immune responses through innate and cellular immunity, and lead to spontaneous healing and resolution. On the other hand, dissemination of the disease occurs more frequently as a result of defective immune response in immunosuppressed individuals [6]. In our case, we tried to exclude the commonest causes of immunosuppression in context of Bangladesh (i.e., HIV, tuberculosis, diabetes mellitus).

There might be multiple risk factors associated with recurrence of VL in immunocompetent patients. A large observational cohort study concluded that male sex, age <5 years and ≥45 years, slower decrease in spleen size ≤0.5 cm at discharge, and shorter duration of symptoms prior to treatment were associated with recurrence of VL [8]. However, a study conducted in Georgia found delay in diagnosis for >90 days, hemoglobin <6 gm/dL and age <1 year were significantly associated with recurrence of the disease [9]. In our case, the diagnosis was delayed for 6 years since symptom onset, which might have played an important role in recurrent relapses of VL despite adequate treatment with conventional therapies.

Although LAmB is the first choice of treatment of VL in Bangladesh, our patient was treated with this agent for three times previously with optimal dosage and duration, even in combination with another potent drug, miltefosine, with no clinical improvement. We treated our case with high dose of LAmB, as LAmB has a concentration-dependent activity against the parasite, where higher concentration of drug results in greater parasite load reduction and longer persistence of elevated drug levels in liver and spleen [10]. Splenectomy has also been proven to improve quality of life with considerable increase in cellular lineage in treatment resistant cases [11]. Our patient presented with severe pancytopenia on admission. So, we planned for a splenectomy in case of persistent pancytopenia and/or severe thrombocytopenia after treatment. However, as the patient's spleen size was significantly reduced and platelet counts were also increasing, she withdrew her consent to undergo splenectomy.

In recurrent VL relapse cases like this, extended or split dose of LAmB is probably not enough for complete cure, which increases the risk of future relapses. Multiple relapses will cause physical as well as economic burden to the patients, as this is a disease of the poor. So, in immunocompromised patients, who present with relapse VL, secondary prophylaxis may be added to achieve complete cure. In our case, alongside her physical anguish, the patient was emotionally distraught due to social isolation since childhood for her chronic illness and pubertal delay. She was subjected to social stigmatization to an extent that compelled her to drop out of school. During her course of treatment, her pubertal delay was addressed with utmost importance. She developed regular menstrual cycle with hormone replacement and secondary sex characteristics ten months after her anti-leishmania treatment. There is a scarcity of evidence on endocrinological effects of visceral leishmaniasis. However, Verde FAL et. al. reported primary adrenal insufficiency, hypoparathyroidism, primary and secondary

thyroid insufficiencies in their study [12]. In our case, this chronic illness might be the contributing factor for her functional hypopituitarism.

Here we have reported an unusual case of recurrent relapse of VL, who was treated successfully with a modified treatment regimen and the patient remained relapse free after ten months of her treatment completion. Higher concentration of LAmB as loading dose, followed by secondary prophylaxis with lower doses of LAmB in combination with oral miltefosine could be used in patients who present with recurrent relapses after conventional and combination therapies.

## Author Contributions

**Conceptualization:** Fazle Rabbi Chowdhury.

**Investigation:** Md Sadiqur Rahman, Sadia Sabah, Md. Abu Rahat, Rajashree Chowdhury, Faria Hossain, Dinesh Mondal, Tahniyah Haq, Md Rafiqul Alam, Fazle Rabbi Chowdhury.

**Supervision:** Md Rafiqul Alam, Fazle Rabbi Chowdhury.

**Visualization:** Tabiha Binte Hannan.

**Writing – original draft:** Tabiha Binte Hannan, Fazle Rabbi Chowdhury.

**Writing – review & editing:** Tabiha Binte Hannan, Zazeba Hossain, Utshab Roy, S. M. Mahbubur Rahman, Shampa Saha, Tahniyah Haq, Md Rafiqul Alam, Fazle Rabbi Chowdhury.

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
