## [Decision Letter · Decision Letter 0]

18 Jan 2024

Dear Dr Hannan,

Thank you very much for submitting your manuscript "Successful treatment of recurrent visceral leishmaniasis relapse in an immunocompetent adult female with functional hypopituitarism in Bangladesh" for consideration at PLOS Neglected Tropical Diseases. As with all papers reviewed by the journal, your manuscript was reviewed by members of the editorial board and by several independent reviewers. The reviewers appreciated the attention to an important topic. Based on the reviews, we are likely to accept this manuscript for publication, providing that you modify the manuscript according to the review recommendations. 

Sincerely,

Walderez O. Dutra, PhD.

Section Editor

Walderez Dutra

Section Editor

Reviewer's Responses to Questions

**Key Review Criteria Required for Acceptance?**

**Methods**

-Are the objectives of the study clearly articulated with a clear testable hypothesis stated?

-Is the study design appropriate to address the stated objectives?

-Is the population clearly described and appropriate for the hypothesis being tested?

-Is the sample size sufficient to ensure adequate power to address the hypothesis being tested?

-Were correct statistical analysis used to support conclusions?

-Are there concerns about ethical or regulatory requirements being met?

Reviewer #1: The objective of the study was clearly articulated with clear language. The paper is well written and fulfills all the necessary points.

Reviewer #2: Methodological data involved in the case study were not adequately provided.

Reviewer #3: Methods are well described.

**Results**

-Does the analysis presented match the analysis plan?

-Are the results clearly and completely presented?

-Are the figures (Tables, Images) of sufficient quality for clarity?

Reviewer #1: Messages are clean and clear. Tables and figures are okay.

Reviewer #2: Although the case presented is relevant, there are not enough results to publish the manuscript in PlosNTD.

Reviewer #3: Results is clear, although improvements in the table format could be addressed.

**Conclusions**

-Are the conclusions supported by the data presented?

-Are the limitations of analysis clearly described?

-Do the authors discuss how these data can be helpful to advance our understanding of the topic under study?

-Is public health relevance addressed?

Reviewer #1: The conclusion highlighted the future importance of this paper.

Reviewer #2: The case study is important for public health, but it does not bring truly new findings.

Reviewer #3: Nevertheless some vagueness due to the lack of retrospective data or knowledge on the endocrinology of visceral leishmaniasis the conclusions are firm and clear.

**Editorial and Data Presentation Modifications?**

Reviewer #1: Accept

Reviewer #2: (No Response)

Reviewer #3: Some improvement in the format of the table.

**Summary and General Comments**

Reviewer #1: I think this article will create enormous consciousness among the clinician because there is clear guideline for treating refractory VL cases.

Reviewer #2: Although the discussion of cases of therapeutic failure in leishmaniasis is of great interest to public health, the case presented in this manuscript failed to adequately discuss the impacts of alternative therapeutic regimens,

Reviewer #3: Excellent article that deserves to be published soon.

PLOS authors have the option to publish the peer review history of their article (what does this mean?). If published, this will include your full peer review and any attached files.

Reviewer #1: Yes: Dr. SHREBASH PAUL

Reviewer #2: No

Reviewer #3: No

Figure Files:

Data Requirements:

Reproducibility:

References

---

## [Decision Letter · Decision Letter 1]

4 Apr 2024

Dear Dr Hannan,

We are pleased to inform you that your manuscript 'Successful treatment of recurrent visceral leishmaniasis relapse in an immunocompetent adult female with functional hypopituitarism in Bangladesh' has been provisionally accepted for publication in PLOS Neglected Tropical Diseases.

Best regards,

Susan Madison-Antenucci, PhD

Section Editor

Walderez Dutra

Section Editor

Reviewer's Responses to Questions

**Key Review Criteria Required for Acceptance?**

**Methods**

-Are the objectives of the study clearly articulated with a clear testable hypothesis stated?

-Is the study design appropriate to address the stated objectives?

-Is the population clearly described and appropriate for the hypothesis being tested?

-Is the sample size sufficient to ensure adequate power to address the hypothesis being tested?

-Were correct statistical analysis used to support conclusions?

-Are there concerns about ethical or regulatory requirements being met?

Reviewer #1: In this version of the manuscript, the objective of the study is clearly articulated. The study design appropriately addresses the objective. Concerns about ethical requirements are appropriately maintained.

Reviewer #2: The methods are suitable for the type of manuscript (case study).

**Results**

-Does the analysis presented match the analysis plan?

-Are the results clearly and completely presented?

-Are the figures (Tables, Images) of sufficient quality for clarity?

Reviewer #1: The case description is now very clear and easy to understand. The tables, figures and discussion are very standard.

Reviewer #2: This is a simple case study, with a topic that is not necessarily new. The authors did not present results that justify publication in PlosNTD.

**Conclusions**

-Are the conclusions supported by the data presented?

-Are the limitations of analysis clearly described?

-Do the authors discuss how these data can be helpful to advance our understanding of the topic under study?

-Is public health relevance addressed?

Reviewer #1: The conclusion contains a very clear message in which the relevance of public health is properly addressed.

Reviewer #2: The manuscript has social relevance, but fails to present news that advances the therapy of recurrent relapses of visceral leishmaniasis.

**Editorial and Data Presentation Modifications?**

Reviewer #1: I am satisfied with the author's response, and I would like to suggest "accept.".

Reviewer #2: N/A

**Summary and General Comments**

Reviewer #1: Bangladesh already received the elimination certificate from WHO for VL. At present, the main challenge is to cure the relapse cases. I think this case will create good awareness amongst the physicians who are dealing with such cases, and it will be a great example for the physicians of other endemic countries.

Reviewer #2: The manuscript does not present enough new data to advance the knowledge already established in leishmaniasis therapies.

PLOS authors have the option to publish the peer review history of their article (what does this mean?). If published, this will include your full peer review and any attached files.

Reviewer #1: **Yes: **Dr. Shrebash Paul

Reviewer #2: No

---

## [Editor Report · Acceptance letter]

12 Apr 2024

Dear Dr Hannan,

We are delighted to inform you that your manuscript, "Successful treatment of recurrent visceral leishmaniasis relapse in an immunocompetent adult female with functional hypopituitarism in Bangladesh," has been formally accepted for publication in PLOS Neglected Tropical Diseases.

Best regards,

Shaden Kamhawi

co-Editor-in-Chief

Paul Brindley

co-Editor-in-Chief
